

# Local-scale topoclimate effects on treeline elevations: a country-wide investigation of New Zealand's southern beech treelines

Bradley S. Case[1] and Hannah L. Buckley[2]

[1] Department of Informatics and Enabling Technologies, Lincoln University, Lincoln, Canterbury, New Zealand
[2] Department of Ecology, Lincoln University, Lincoln, Canterbury, New Zealand

## ABSTRACT

Although treeline elevations are limited globally by growing season temperature, at regional scales treelines frequently deviate below their climatic limit. The cause of these deviations relate to a host of climatic, disturbance, and geomorphic factors that operate at multiple scales. The ability to disentangle the relative effects of these factors is currently hampered by the lack of reliable topoclimatic data, which describe how regional climatic characteristics are modified by topographic effects in mountain areas. In this study we present an analysis of the combined effects of local- and regional-scale factors on southern beech treeline elevation variability at 28 study areas across New Zealand. We apply a mesoscale atmospheric model to generate local-scale (200 m) meteorological data at these treelines and, from these data, we derive a set of topoclimatic indices that reflect possible detrimental and ameliorative influences on tree physiological functioning. Principal components analysis of meteorological data revealed geographic structure in how study areas were situated in multivariate space along gradients of topoclimate. Random forest and conditional inference tree modelling enabled us to tease apart the relative effects of 17 explanatory factors on local-scale treeline elevation variability. Overall, modelling explained about 50% of the variation in treeline elevation variability across the 28 study areas, with local landform and topoclimatic effects generally outweighing those from regional-scale factors across the 28 study areas. Further, the nature of the relationships between treeline elevation variability and the explanatory variables were complex, frequently non-linear, and consistent with the treeline literature. To our knowledge, this is the first study where model-generated meteorological data, and derived topoclimatic indices, have been developed and applied to explain treeline variation. Our results demonstrate the potential of such an approach for ecological research in mountainous environments.

Corresponding author
Bradley S. Case,
Bradley.Case@lincoln.ac.nz

## INTRODUCTION

The tree limit, the uppermost elevation at which trees can survive, is a global bioclimatic phenomenon ultimately determined by growing season temperature (*Körner & Paulsen, 2004*). However, treelines can often deviate below this potential climatic treeline, due to the influence of factors that operate at multiple spatial scales (*Holtmeier & Broll, 2005*; *Malanson et al., 2011*; *Case & Duncan, 2014*). These factors include: differences in regional climate regime (heat and moisture) that affect tree growth and mediate finer-scale influences on treeline (*Daniels & Veblen, 2003*); disturbances, such as avalanches and landslides (*Daniels & Veblen, 2003*; *Leonelli, Pelfini & Cella, 2009*; *Case & Hale, 2015*); spatial variation in the distribution of moisture and nutrients related to geomorphology (*Butler et al., 2007*), and local-scale climatic variability related to topography ('topoclimate') (*Holtmeier & Broll, 2005*). Thus, there is a growing recognition that in order to understand how treelines may respond regionally to climatic change, datasets and methods will be required for characterising and modelling influences on treelines at a range of scales and across large areas (*Holtmeier & Broll, 2007*).

It is relatively easy to obtain GIS-ready datasets, such as gridded climatic and topographic data, for investigating influences on treelines at a regional scale (e.g., *Case & Hale, 2015*). However, spatially-explicit local scale data, particularly for climatic variables, are typically more challenging to obtain. One solution is to employ topographic indices such as slope, aspect, and terrain shape, derived in a GIS from digital elevation models (DEMs), that can act as proxies for the effects of local-scale variation in environmental conditions (*Moore, Grayson & Ladson, 1991*). A number of treeline studies have used DEM-derived indices in this way to highlight the important role that topography plays in influencing treeline variability (*Brown, 1994*; *Allen & Walsh, 1996*; *Walsh et al., 2003*; *Dullinger, Dirnböck & Grabherr, 2004*; *Bader & Ruijten, 2008*). For example, *Brown (1994)* used three DEM-derived topographic characteristics to explain the presence of four treeline transition vegetation types. Similarly, *Bader & Ruijten (2008)* found that a DEM-derived topographic index was a significant factor in explaining the presence and absence of forest within the treeline zone; the index described convex landscape zones where cold air drainage occurred and caused inverted treelines. Thus DEM-derived indices have proven useful for gaining important insights into treeline variation in cases where direct measures of local climate, disturbance and other variables have not been available.

However, there are limits to the extent to which local-scale topoclimatic effects, in particular, can be sufficiently represented by such DEM-derived indices. Daily variation in local wind speeds, for instance, are the result of a range of meteorological processes inducing effects such as valley and downslope winds, cold air ponding, and differential irradiation, which are highly variable in space and time (*Daly, Conklin & Unsworth, 2010*). Such topoclimatic phenomena are likely to have significant impacts on treelines because they can either ameliorate or exacerbate physiological functioning at the plant scale, which can have knock-on effects for recruitment above treeline (*Rehm & Feeley, 2015*). Therefore, new approaches that can produce reliable topoclimatic data for local-scale treeline studies are warranted.

One such approach is to use predictive, numerical climate models that are capable of generating accurate estimates of meteorological parameters in complex terrain and that can be applied to different sites without the need for local parameterisation. A number of readily-available mesoscale atmospheric models are suited to this task. For example, The Air Pollution Model (TAPM) produced by CSIRO Australia (*Hurley, 2008a*) is a mesoscale model that has been applied at sites worldwide (*Hurley, Edwards & Luhar, 2008*) and has been shown to be able to account for topographically-mediated meteorological processes such as cold air drainage and ponding in complex terrain (*Hurley, Physick & Luhar, 2005*; *Mocioaca, Sivertsen & Cuculeanu, 2009*; *Case, Zawar-Reza & Tait, 2015*). *Case, Zawar-Reza & Tait (2015)* used TAPM to generate spatially-explicit meteorological data for a range of sites across New Zealand and showed that TAPM could relatively accurately simulate temperature and wind speed at these sites and that prediction accuracies were relatively consistent among sites and years for the different variables examined.

Daily and monthly variation in wind speed, temperature extremes, solar radiation, and relative humidity, and interactions among these variables, together define possible topoclimatic conditions at the local treeline. These meteorological variables rarely affect treelines in isolation, but rather work synergistically to produce conditions that affect trees' physiological performance. For instance, although high winds can potentially cause direct physical damage to trees at high altitudes, this type of damage alone is typically not a critical factor in explaining treeline formation (*Körner, 1998*). More damaging, however, is when the action of wind combines with other topoclimatic variables to produce cumulative stressful conditions for trees over time. Such an example might be when high winds, together with high temperatures and low relative humidity during hot summer months, produce conditions where desiccation stress is more likely to occur (*Köhler, Gieger & Leuschner, 2006*; *Moyes et al., 2013*). Similarly, while low night time temperatures on their own will likely have little impact on seedlings at treeline, when combined with low windspeeds and high amounts of outgoing radiation, frosts can occur that can affect leaves and buds, especially early in the growing season (*Jordan & Smith, 1994*). There are also potential positive effects: for example, locations that generally have higher warmth and higher inputs of sunlight, in the absence of other stressors, might be expected to have conditions more suitable for tree establishment and growth (*Cairns & Malanson, 1998*). Hence, research that is able to explore the relevance of these combined effects across different locations will be able to provide new insights into the importance of topoclimate in determining local treeline variability, relative to other local and regional influences.

In this study we provide an assessment of the importance of topoclimatic factors, relative to landform and regional environmental factors, in determining local-scale variations in treeline position for c. 2,100 treeline locations at 28 study areas across New Zealand. We focus here on southern beech (Nothofagaceae) treelines, which comprise the majority of New Zealand's treeline zones. These treelines are highly-abrupt and have had minimal above-treeline recruitment of new stems over the past 20 or more years (*Harsch et al., 2012*). Thus, while we acknowledge that demographic processes ultimately drive treeline formation at a given site, this study focusses specifically on among-site

spatial variation in treeline elevation and the relative roles of topoclimate *versus* other factors in describing such variation at this spatial scale. Previous studies have highlighted the considerable variability observed in the elevation of New Zealand's southern beech (Nothofagaceae) treelines both locally and regionally, and the possible explanations for this variability (*Wardle, 2008*; *Case & Duncan, 2014*; *Case & Hale, 2015*). For example, an analysis by *Case & Duncan (2014)* indicated that the position of treeline varied mainly due to solar radiation and mountain mass effects at a range of scales across the country, although the coarseness of the explanatory data limited the degree to which local-scale effects could be reliably assessed. Based on field observations, *Wardle (1985a)*, *Wardle (1985b)*, *Wardle (1985c)* and *Wardle (2008)* posited that local-scale variations in beech treeline elevation are related to differences in landform at treeline, with treelines reaching higher elevations on steep slopes and convex curvatures than on gentler concave forms, although the pervasiveness of this pattern across the country has not yet been empirically evaluated. We compile an explanatory dataset comprising regional-scale climate, disturbance, and landscape variability factors, local-scale DEM-derived landform factors, and a set of novel topoclimatic indices derived from meteorological data generated using the TAPM meso-scale atmospheric model (*Case, Zawar-Reza & Tait, 2015*). With these data we address two main questions: (1) Are treelines at different locations across New Zealand characterised by distinctive topoclimatic conditions?; and (2) What is the nature and extent of the effect of topoclimatic stress on the variability in treeline elevation among sample points, relative to landform and regional drivers?

## METHODS

### Study areas and treeline delineation

We used a GIS-based dataset delineating southern beech treelines in New Zealand (*Case & Duncan, 2014*). Given the abruptness of these treelines, we used available landcover data to easily delineate treeline boundaries as the polygon boundaries between the "Indigenous Forest" landcover class and four adjacent subalpine landcover classes (see *Case & Duncan, 2014* for details). Once identified, these treeline boundaries were extracted as line features in the GIS and points were generated along these treelines at an average spacing of approximately 1 km in order to capture local scale variability. These points formed the basic unit for extracting the elevation, meteorological and landform data at treeline that were used for subsequent analyses. Next, we chose 28 treeline study areas across the country as a basis for atmospheric modelling with the TAPM model (Fig. 1). Study areas were $7 \times 7$-km ($49$-km$^2$) in size and were randomly located across southern beech treeline zones from approximately 46°S latitude in the south of the country to 39°S latitude in the north. Study area dimensions were determined by the requirements of the TAPM model and its application for our research aims (see below). The mean distance from each site location to the next closest site was 32.4-km. To verify that all treeline point locations within these study areas were actually located at treeline, we visually assessed sample points against georeferenced, 15-m resolution, SPOT 5 satellite imagery (Fig. 1, inset). Points that were not within 50-m of the treeline seen on the imagery were manually re-positioned to
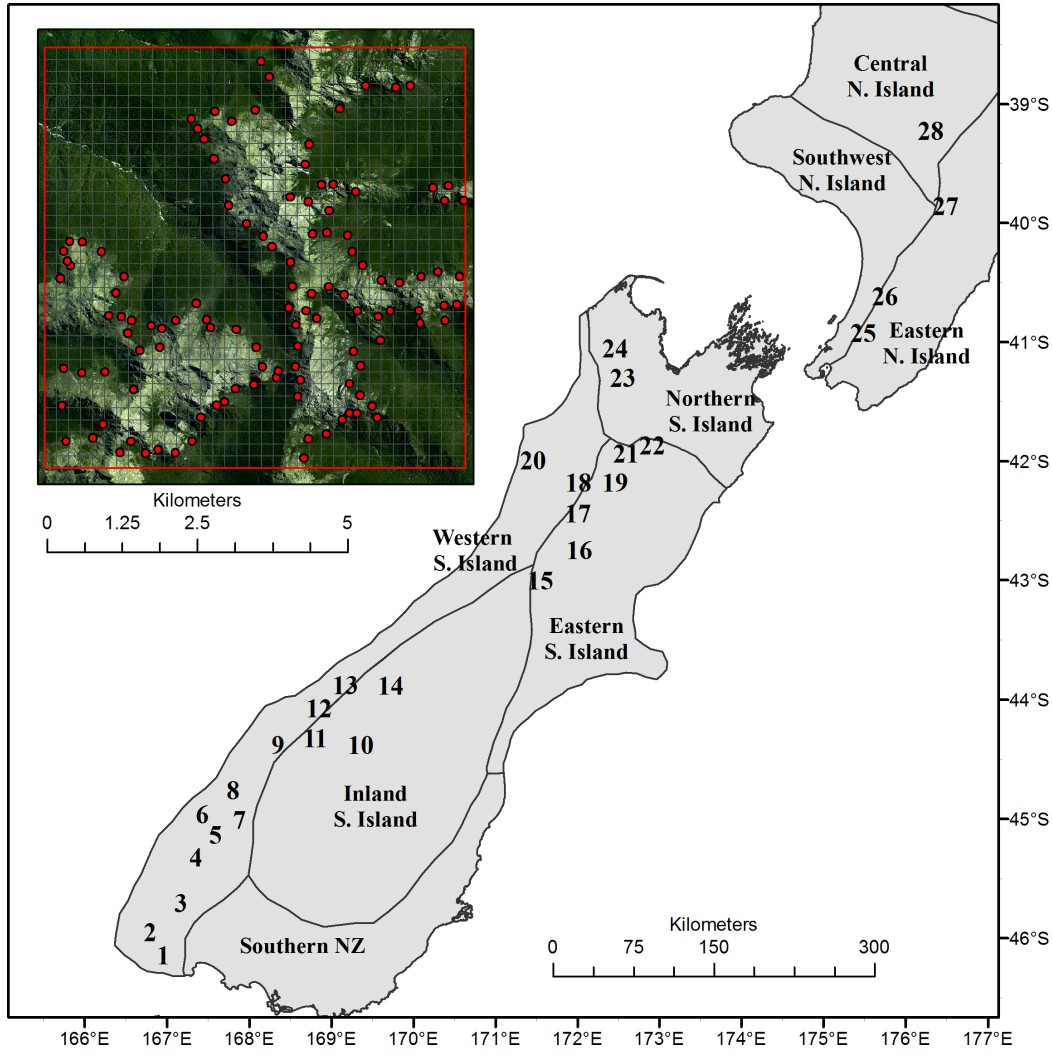

**Figure 1** **Location of the 28 study sites across New Zealand relative to broad climatic regions.** The inset (top left) shows one study area with treeline locations used for analyses, and the 200 m resolution, 7 × 7 km grid across which meteorological outputs are generated by the TAPM model. In the background of the inset is an example of a SPOT 5 satellite image used as a basis to verify that treeline points were accurately located at treeline.

the nearest treeline edge; those that could not be verified as being at treeline due to the presence of shadow or cloud in the imagery, were removed from the dataset. This process resulted in a total of 2,189 points located at treeline across the 28 study areas.

## Datasets

### Treeline elevation data

At each study area, we assumed that the treeline observation occurring at the highest elevation provided a reasonable index of the potential, climatically-driven treeline in that locale. Thus, in the absence of disturbances and topoclimatic stressors, all treelines in the vicinity should reach this maximum observed local elevation due to the theoretical

dominant effect of mean growing season temperature (*Körner & Paulsen, 2004*). Based on this assumption, the difference in observed treeline elevations at each sample location from their site-level maximum ("elevation deviation from maximum") was computed and used as the response variable in statistical analyses. This response variable also provided a standardised measure of treeline variability across the 28 study areas in that it removed possible confounding effects due to the negative trend between treeline elevation and latitude that occurs across New Zealand. To calculate elevation deviations, treeline elevation at each location was extracted in the GIS from a 25-m resolution digital elevation model for New Zealand (*Barringer, McNeill & Pairman, 2002*) and then subtracted from the maximum observed treeline elevation across all locations within each study area.

### Landform variability data

Two variables, slope gradient and surface curvature, were derived from DEM data to investigate the impact of landform on treeline elevation deviation. Percentage slope gradient was derived from DEM data at each location using the "Slope" function within ArcGIS 10.1. The slope gradient is calculated as the rate of maximum change in elevation among a 3 × 3 neighbourhood of DEM cells surrounding a focal cell location. The degree of convexity or concavity of the landsurface was derived from DEM data at each treeline location using the "curvature" function within ArcGIS 10.1. This function determines surface curvature for each cell of the DEM by fitting a fourth order polynomial to the elevation surface within a 3 × 3 moving window centred around a given cell location. The resulting value from this calculation is either positive, signifying a convex shape, or negative signifying a concave shape (*Zevenbergen & Thorne, 1987*).

### Regional-scale environmental data

Mean values for growing season temperature, mountain mass, precipitation, and earthquake intensity were extracted for each 7 × 7-km study area to represent these potential among-site differences across New Zealand. Growing season temperature and precipitation data for the study areas were extracted from 500-m resolution gridded climate layers for New Zealand (*Wratt et al., 2006*). A mountain mass index, which represents the effect of mountain size on the regional thermal regime, was derived in the GIS by determining the amount of area above 1,200-m within each of the study site zones. Earthquake data were extracted from a 500-m resolution spatial dataset of the expected mean peak ground acceleration within a 150 year return interval, expressed as the proportion of the acceleration due to gravity (*Stirling, Mc Verry & Berryman, 2002*).

### TAPM-generated meteorological data

The Air Pollution Model (TAPM) V.4 (*Hurley, 2008a*) was used to generate meteorological data at the 28 study sites for the purpose of characterising topoclimatic conditions. TAPM predicts three-dimensional meteorology at scales ranging from relatively coarse (1,000 to 1,500-km) to fine (<500-m) (see *Hurley, Physick & Luhar, 2005*; *Hurley, 2008b* for further details regarding TAPM). For this study, we ran TAPM for both January and July 2002 at the 28 study areas because the treeline data were also from 2002, and because we wanted to account for both winter and summer conditions in assessing topoclimatic

effects. Further, a previous study at these same sites in New Zealand (*Case, Zawar-Reza & Tait, 2015*) showed that TAPM outputs for 2002 were consistent for the overall period of 2001–2007 and, on average, did not differ from 30-year climate normal data for maximum and minimum temperature and wind speed when evaluated across all sites. Thus, we considered 2002 TAPM data to be representative, particularly for exploring among-site topoclimatic variation.

To run TAPM at its finest resolution (200-m grid cells), square $7 \times 7$-km study areas centred on each site location were established, resulting in the generation of meteorological estimates for 1,225 grid cells at each study area for each of the two months. Meteorological outputs from the model comprised hourly data for screen-level (2-m) air temperatures (°C), relative humidities (%), and short- and long-wave radiation values (Watts m$^{-2}$), and for wind speeds at 10 m above the ground (m s$^{-1}$). To obtain TAPM-generated data at each treeline sample point, we first generated geo-referenced grids of data for each topoclimatic variable within ArcGIS at each site. We then extracted these data to treeline point locations in the GIS using standard raster-to-point data extraction methods.

## Derivation of topoclimatic indices

Using TAPM-simulated data, we computed five topoclimatic indices that provided relative, potential measures of topoclimatic stress (indices of photoinhibition, desiccation, and frost), and topoclimatic amelioration (insolation) on tree physiological function at treeline. All indices were computed for the months of January and July in order to determine how these indices vary in summer and winter, and if this variation is important for understanding treeline variability.

### *Photoinhibition index*

There is considerable evidence from treeline research that increased sky exposure is detrimental to seedling establishment above existing treelines (*Wardle, 1985b*; *Ball, Hodges & Laughlin, 1991*; *Germino & Smith, 1999*; *Germino, Smith & Resor, 2002*; *Bader, Geloof & Rietkerk, 2007*; *Giménez-Benavides, Escudero & Iriondo, 2007*). This effect is typically attributed to cold-induced photoinhibition, where low temperatures and high solar radiation combine to disrupt photosynthetic functioning. To generate an index of photoinhibition, hourly temperature values ($T_{\text{hourly}}$) were first rescaled relative to the overall observed site-level maximum temperature such that lower temperatures received a higher relative weighting in calculating the potential for photoinhibition at a given treeline location. The photoinhibition index was then calculated as the product of rescaled hourly temperatures and hourly solar radiation ($\text{SolRad}_{\text{hourly}}$) summed across all daytime hours and divided by the number of monthly daytime hours ($N_{\text{daytime}}$):

$$\text{Mean photoinhibition index} = \frac{\sum_{\text{daytime}}[(T_{\text{site\_max}} - T_{\text{hourly}}) \times \text{SolRad}_{\text{hourly}}]}{N_{\text{daytime}}}. \quad (1)$$

Thus, locations where the combination of low temperatures and high solar radiation values were more prevalent received higher values for this index.

### Summer desiccation index

Desiccation stress can occur at treeline in both summer and winter (*Tranquillini, 1979*; *Harsch & Bader, 2011*). During hot summer months, desiccation conditions can result from the combination of relatively high daytime wind speeds and temperatures and low relative humidities. These conditions can increase cuticular transpiration rates and dry out thin soils, leading to drought stress (*Cui & Smith, 1991*; *Kullman, 2005*). In winter months, cold temperatures periodically induce frozen soils and plant tissues, while high wind conditions and low relative humidities increase cuticular transpiration rates, thus increasing the potential for desiccation damage of treeline seedlings and trees (*Baig & Tranquillini, 1980*; *Wardle, 1981*; *Sowell, McNulty & Schilling, 1996*). High wind speeds can also exacerbate these effects by causing direct abrasion damage to the cuticles of exposed leaves, thereby increasing the potential for water loss (*Hadley & Smith, 1983*). Indices of potential summer and winter desiccation conditions were calculated as the mean product of daytime hourly temperature, wind speed and relative humidity data for January. For both summer and winter desiccation indices, relative humidities were rescaled relative to a maximum humidity of 100% such that lower humidity values contributed towards a higher desiccation index:

Mean summer desiccation index

$$= \frac{\sum_{\text{daytime}} [T_{\text{hourly}} \times \text{WS}_{\text{hourly}} \times (100 - \text{RelHum}_{\text{hourly}})]}{N_{\text{daytime}}}. \tag{2}$$

### Winter desiccation index

This index was similarly calculated as in (2), but averaged across the whole day (i.e., total number of monthly hours, $N_{\text{total}}$) and with temperatures rescaled such that low temperatures (inducing possible frost drought conditions) contribute more to high index values:

Mean winter desiccation index

$$= \frac{\sum_{\text{daytime}} [(T_{\text{site\_max}} - T_{\text{hourly}}) \times \text{WS}_{\text{hourly}} \times (100 - \text{RelHum}_{\text{hourly}})]}{N_{\text{total}}}. \tag{3}$$

### Frost index

Cold night-time temperatures combined with low wind speeds and high levels of outgoing, long-wave radiation can lead to frost conditions (*Lindkvist, Gustavsson & Bogren, 2000*). Frost damage to mature trees at treeline is rarely significant and is not considered a major treeline-forming factor (*Körner, 1998*; *Cieraad et al., 2012*). However, early summer frosts can cause significant damage to new leaves of seedlings (*King & Ball, 1998*) and is considered a potential limiting factor for the establishment of trees above existing treelines in certain regions of the world (*Germino & Smith, 2000*; *Piper et al., 2005*) including *Nothofagus* treelines in New Zealand (*Wardle, 1985a*; *Greer & Buxton, 1989*). A frost index was computed as:

Frost index

$$= \frac{\sum_{\text{nighttime}}[(T_{\text{site\_max}} - T_{\text{hourly}}) \times (\text{WS}_{\text{site\_max}} - \text{WS}_{\text{hourly}}) \times (-(\text{NetRad}_{\text{hourly}})]}{N_{\text{nighttime}}}. \quad (4)$$

$T_{\text{site\_max}}$ and $\text{WS}_{\text{site\_max}}$ are the maximum, site-level temperature and wind speed values, and are used to rescale hourly temperatures ($T_{\text{hourly}}$) and wind speeds ($\text{WS}_{\text{hourly}}$) to a reverse scale at each site. In this way, temperatures and wind speeds that are low relative to the site-level maximum for these variables contribute towards a higher frost index value, while those that approach the site-level maximum contribute towards a lower frost index. $\text{NetRad}_{\text{hourly}}$ is the hourly net radiation, computed by TAPM as the difference between incoming solar radiation and outgoing radiation emitted from the land surface; $\text{NetRad}_{\text{hourly}}$ values during night-time hours are negative as there is no incoming solar radiation. Hourly frost index values calculated in this manner are then summed across all night-time hours in each of January and July, and divided by the number of monthly night-time hours for those months ($N_{\text{night-time}}$).

### Insolation

Differences in solar radiation loadings among treeline locations typically reflect either differences in topographic orientation relative to the sun (i.e., aspect differences) or differences in the amount of cloud cover over time. Insolation is a key topoclimatic variable at treeline, and can exert both positive and negative effects. In general, locations with higher solar radiation, that are not also subjected to cold night-time/early morning temperatures, are likely to have more favourable conditions for growth due to increased warmth (*Danby & Hik, 2007a*). Insolation values were computed as the total daytime solar radiation at a location.

## Data analysis

We used principal components analysis (PCA) to decompose variation in, and examine correlations among, the raw, TAPM-generated topoclimatic variables across the 28 study areas. To visualise these outputs, we plotted PCA bi-plots of axis combinations that explained at least 10% of the variation in the multivariate data; the location of study areas in ordination space were also overlain onto these bi-plots and coloured by their latitudes, to determine if topoclimatic data showed broad geographic structuring.

To investigate relationships between the response variable and the 17 explanatory factors, we used a two-stage approach: first we used random forest analysis (*Breiman, 2001*) to estimate and rank the importance of each explanatory factor in describing variability in the response variable; second, we used conditional inference trees to gain further insight into the nature of relationships between the response variable and most important explanatory factors. These two non-parametric, machine learning analysis methods have several benefits over linear parametric modelling methods in that they can more easily model complicated, non-linear relationships among large numbers of inter-correlated explanatory factors and a response variable (*De'ath & Fabricius, 2000*; *Cutler et al., 2007*). The random forest approach is an ensemble machine learning method

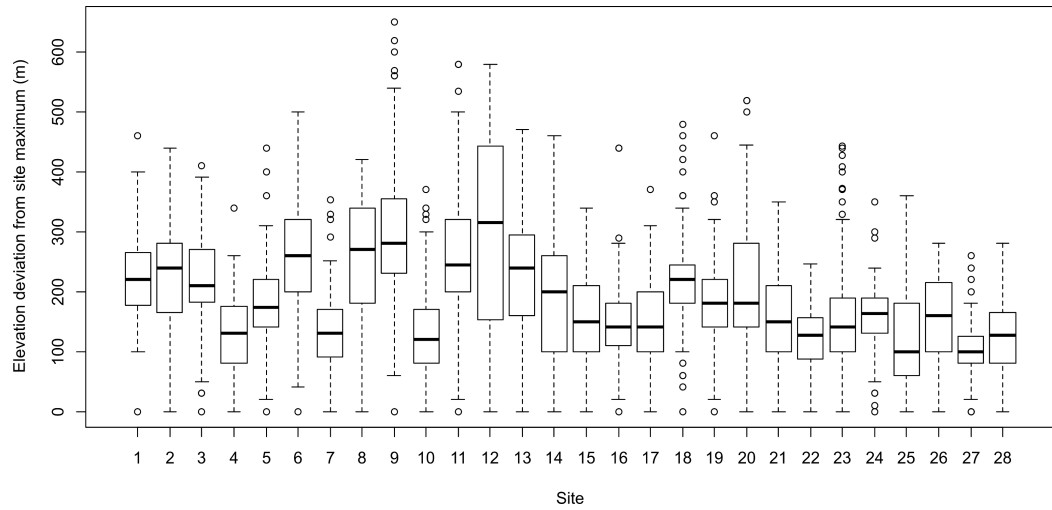

**Figure 2** The distribution of treeline elevation deviations (m) from their study area maxima for each of the 28 study areas examined in the study. The deviation values therefore reflect the degree to which treeline elevations are lower than the site maximum (i.e., where the elevation deviation equals zero).

that averages the outcomes of thousands of boostrapped regression trees ('forests') in order to generate a relative ranking of the importance of the explanatory variables in predicting the response variable across these forests. The random forest algorithm, its associated metrics, and uses in ecology are detailed in *Cutler et al. (2007)*. We used the random forest 'variable importance' measure to identify the most influential factors in explaining variation in the response variable and then used partial dependence plots to show the marginal effect of each of these factors on the response variable while holding all of the other explanatory factors at their average values (*Cutler et al., 2007*). The relationships of top-ranked explanatory factors with the response variable were then further investigated using conditional inference trees (*Hothorn, Hornik & Zeileis, 2006*), a recursive partitioning regression tree method that generates a set of decision rules describing how variation in the response data is best partitioned in terms of the explanatory data. The conditional inference tree method requires a statistically significant difference, as determined by Monte Carlo simulation, in order to create a partition in the data; this algorithm, in comparison to those used by other regression tree methods, minimises bias and prevents over-fitting and the need for tree pruning (*Hothorn, Hornik & Zeileis, 2006*). Random forest and conditional inference tree analyses were implemented in R version 3.1.0 using the 'randomForest' (*Liaw & Wiener, 2002*) and 'party' (*Hothorn, Hornik & Zeileis, 2006*) packages, respectively.

## RESULTS
Treeline elevation across the 28 study areas ranged from 763 to 1,486-m (mean elevation = 1,112.8-m, standard deviation = 195.5-m). There was a large amount of variation within and among study areas in the deviation of treeline elevations from their potential study area maxima (Fig. 2). Across study areas, treeline elevations deviated from near the study area maximum (i.e., 0-m elevation deviation) to 500 m lower in elevation at some

**Table 1** **Table of PCA loadings.** Axis loadings for a principal components analysis of January and July hourly meterological data generated by the TAPM model across treeline locations in this study. Loadings greater or equal to 0.25 are shown in bold.

| TAPM-generated meteorological variable | PC1 | PC2 | PC3 |
|---|---|---|---|
| January net outgoing long-wave radiation | **−0.27** | 0.03 | −0.17 |
| January total solar radiation | **0.25** | −0.08 | 0.06 |
| July net outgoing long-wave radiation | **−0.35** | −0.03 | 0.11 |
| July total solar radiation | **0.28** | −0.17 | −0.11 |
| January rain days | **−0.29** | −0.14 | −0.02 |
| July rain days | **−0.31** | −0.13 | 0.16 |
| July minimum relative humidity | **−0.32** | −0.06 | 0.02 |
| July maximum relative humidity | **−0.31** | −0.14 | −0.04 |
| January minimum relative humidity | **−0.27** | **−0.23** | −0.12 |
| January maximum relative humidity | **−0.26** | **−0.27** | −0.15 |
| January minimum temperature | −0.14 | **0.38** | 0.03 |
| January maximum temperature | −0.02 | **0.38** | −0.03 |
| July minimum temperature | −0.14 | **0.39** | 0.05 |
| July maximum temperature | −0.03 | **0.42** | −0.06 |
| January frost hours | 0.21 | **−0.27** | −0.05 |
| July frost hours | 0.14 | **−0.22** | **−0.36** |
| January minimum wind speed | 0.13 | −0.05 | **0.41** |
| January maximum wind speed | 0.12 | −0.03 | **0.44** |
| July minimum wind speed | −0.07 | −0.14 | **0.43** |
| July maximum wind speed | −0.06 | −0.12 | **0.43** |

locations. For the majority of treeline locations, elevations were predominately in the range of 100–250 m lower than the maximum potential treeline.

Approximately 80% of variation in the meteorological data generated by TAPM across the treeline locations was explained by the first three principal component axes. The first principal component axis was most strongly characterised by negative loadings for January and July minimum and maximum relative humidity, rain days, and night time longwave radiation, and positive loadings for January and July solar radiation (Table 1). The second principal component axis was associated with relatively high positive loadings for January and July maximum and minimum temperatures and negative loadings for January frost hours. The third principal component axis was most strongly characterised by positive loadings for January and July minimum and maximum wind speeds and negative loadings for July frost hours. Study areas were well-separated along these gradients and there were geographic patterns in the positioning of study areas in multivariate space, although these patterns were not obviously latitudinally-driven (Fig. 3). Rather, sites were clearly separated based on dominant regional differences in mountain climates in terms of warmth, moisture, windiness, solar radiation, and frostiness.

The random forest analysis explained approximately 50% of variation in treeline elevation deviation based on the 17 explanatory factors (see Supplemental Information 1 for summary statistics). Variable importance rankings indicated that the landform variable "curvature" had the largest impact in explaining variation in treeline elevation

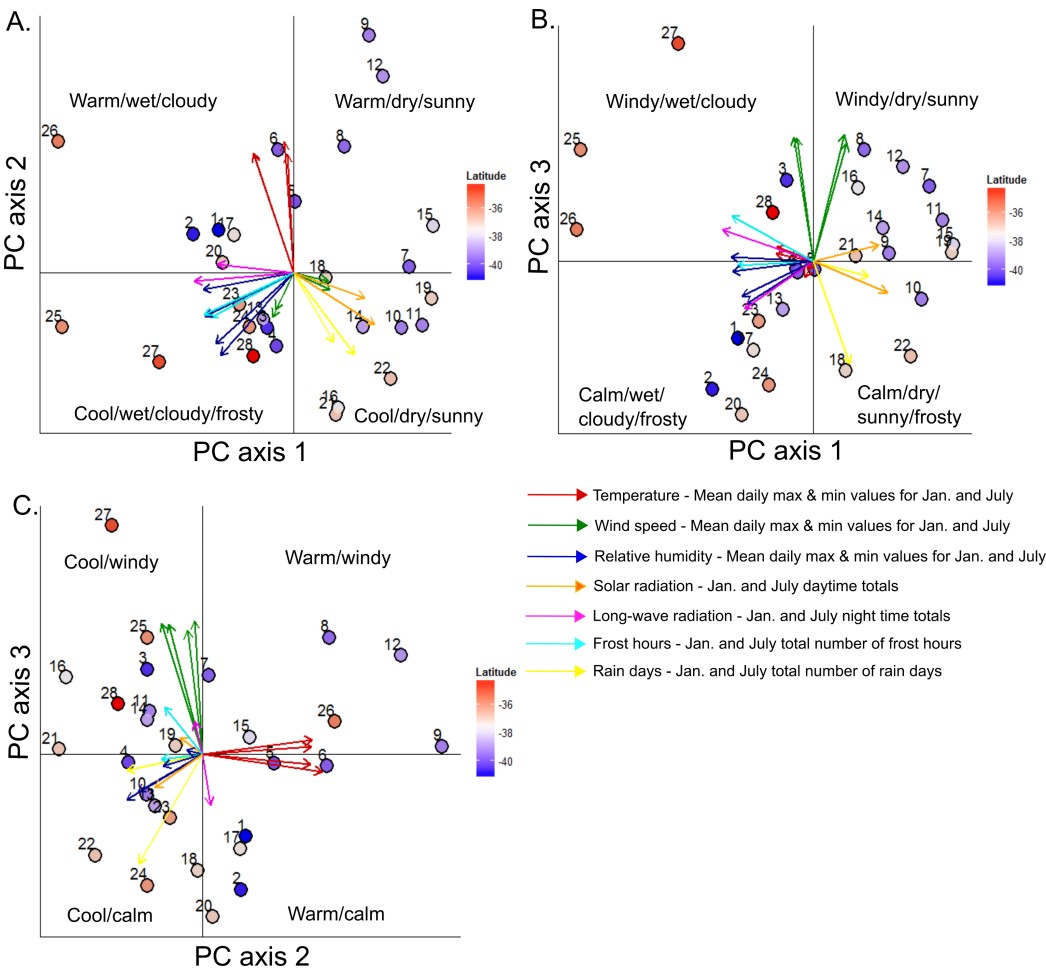

**Figure 3** Bi-plots of a principal components analysis (PCA) of meterological data generated at southern beech treelines using the TAPM mesoscale atmospheric model. Bi-plots are shown for principal components 1 and 2 (A), 1 and 3 (B), and 2 and 3 (C), which collectively explain 80.2% of the variation in the data. PCA eigenvectors are coloured according to the type of meteorological variable; multiple vectors of the same colour occur when both maximum and minimum conditions for that variable, for both January and July, are included. Dots represent positions of the 28 study areas in ordination space, labelled with the site numbers and coloured by their latitidinal position. See Table 1 for axis loadings for each variable.

deviation across all of the regression trees built by the random forest analysis (Fig. 4). Next, and essentially equivalent in importance, were a group of eleven variables including slope, precipitation, growing season temperature, and all of the topoclimatic indices. The remaining five regional variables—mountain mass index, erosion index, earthquake intensity, and winter temperatures—were clearly ranked lower in importance by the random forest algorithm; further analyses were therefore focussed on the 12 top-ranked variables. Partial dependence plots suggested that relationships between treeline elevation deviation and the top 12 variables were frequently non-linear (Fig. 5). Deviations of treelines from their potential site-level maxima were negatively associated with slope curvature, with treeline elevations deviating most greatly on increasingly concave slopes.

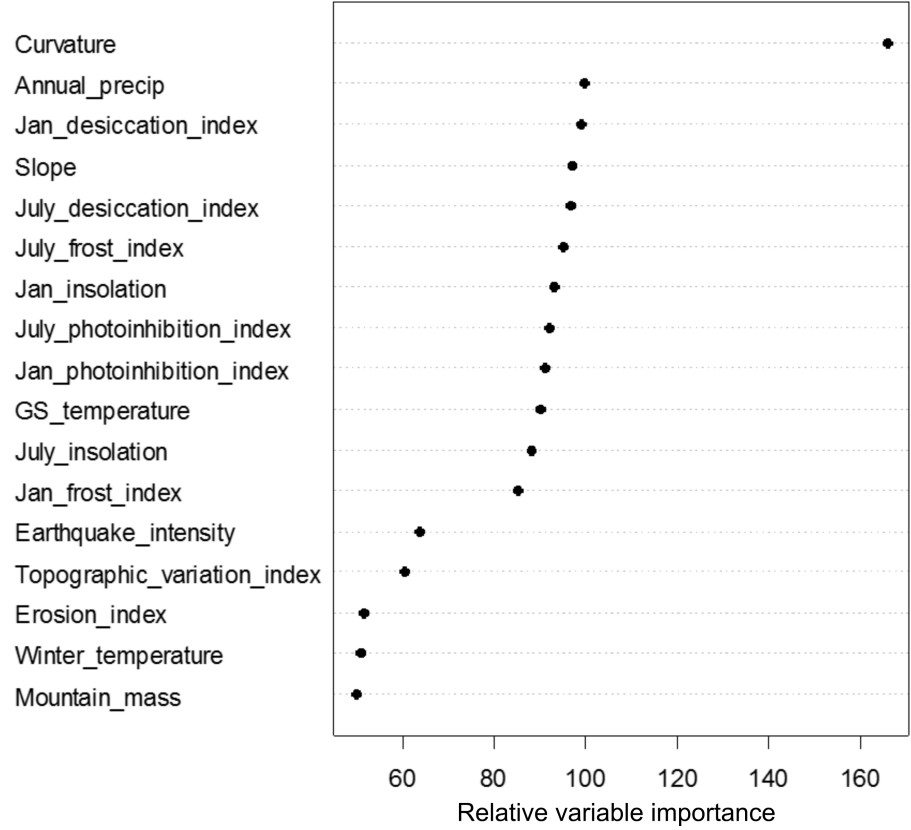

**Figure 4 Variable importance plot resulting from the random forest analysis of the effects of 17 explanatory factors on treeline elevation deviation across the 2,100 treeline locations. The relative importance score reflects the percent increase in mean square error that would result from the removal of a given factor from the analysis.**

Greater deviations in treeline elevation were associated with lower, gentle slope gradients, as well as with steep slopes, with the lowest deviations occurring on slopes of intermediate steepness of about 50–100%. Treelines tended to deviate more from their potential maxima with increasing January and July desiccation index values. Five topoclimatic indices (January and July photoinhibition and frost indices and July insolation) showed a more u-shaped relationship with treeline elevation deviation, with relatively high deviations occurring for both very low index values and higher index values and the lowest deviations at intermediate index values. January insolation showed a negative linear association with treeline elevation deviation. Annual precipitation was generally positively related to treeline deviation, with the highest deviations associated with the most regionally-wet regions. Growing season temperature showed a negative sigmoidal relationship with treeline deviation, with a sharp decrease from high to low treeline elevation deviation occurring at intermediate growing season temperatures of about 11.5–12.5 °C.

The conditional inference tree analysis produced twelve terminal nodes using six of the 12 explanatory variables entered into the analysis: precipitation, curvature, slope gradient, growing season temperature, July frost index, and July desiccation index (Fig. 6). The first

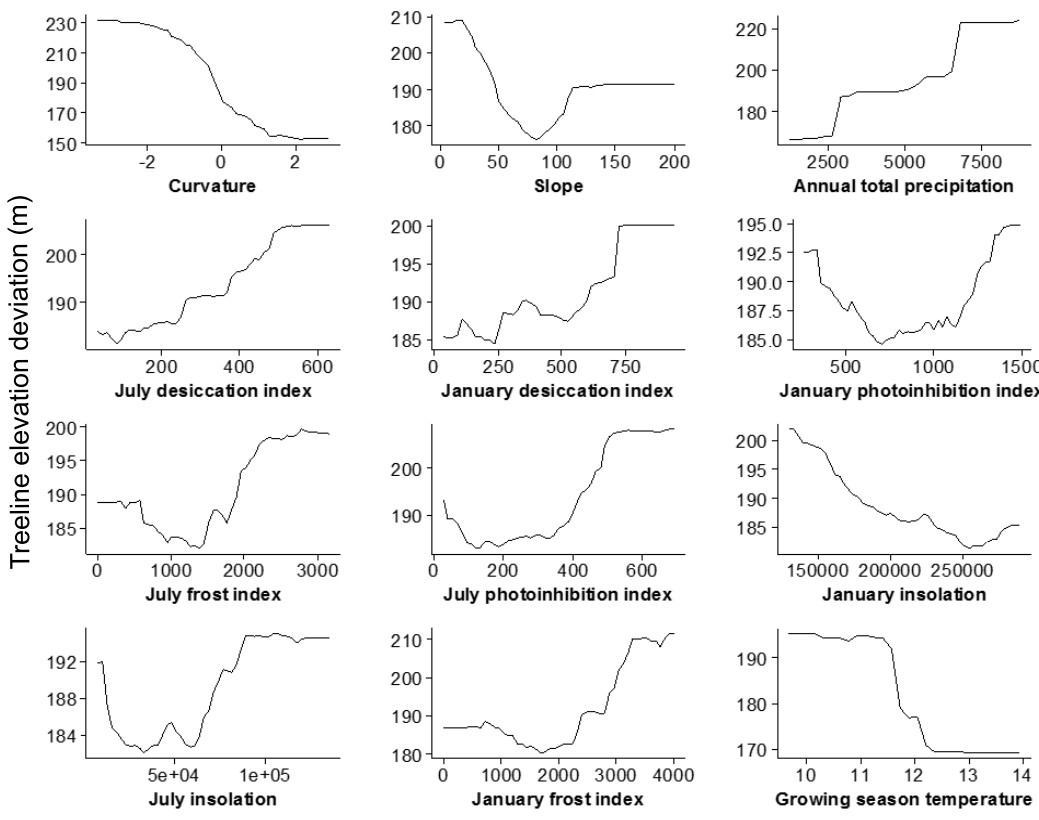

**Figure 5 Partial dependence plots, based on results from the random forest analysis, showing the mean marginal influence of 12 explanatory variables on treeline elevation deviation. Each plot represents the effect of each variable while holding the other variables constant.**

partition of the response dataset was based on a regional annual precipitation threshold. In general, treeline locations with greater than 5,664 mm per year of precipitation showed on average higher deviations from their site-level maxima than locations with precipitation values lower than this threshold. Both the highest and lowest treeline elevation deviation locations were explained by data partitions involving interactions among precipitation, growing season temperature, curvature and July frost index (Fig. 6). While there was some within-site variability in the terminal nodes associated with each study area, there were also clear among-study area regional patterns in the location of terminal nodes (Fig. 7), similar to what emerged from the principal components analysis. For instance, nodes 19–23, representing locations with the highest average treeline deviations, were mainly associated with study areas 6, 8, 9, 12, and 13 which are situated along the wetter, western side of the main divide in the South Island of New Zealand (see Fig. 1). Conversely, treeline locations closest to their potential maxima (node 13) were primarily associated with study areas in the North Island (25–28) and the very top of the South Island (22 and 24).

## DISCUSSION

Although a range of previous treeline studies have used combinations of topography-based metrics and/or interpolated climatic datasets for treeline modelling (e.g., *Brown, 1994*;

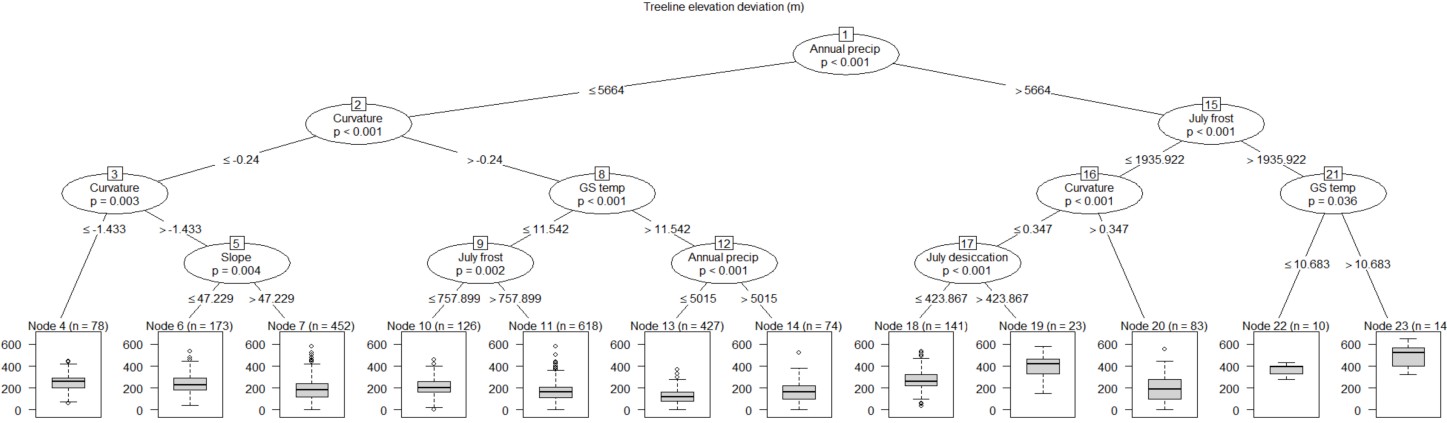

**Figure 6 Conditional inference tree explaining treeline elevation deviation across 28 study areas using the 12 most important explanatory factors determined from the random forest analysis.** The tree shows pathways of how the response data were recursively partitioned based on explanatory variables. The observations associated with each terminal node are the result of these partitionings. For example, Node 4 comprises treeline locations with a median treeline elevation deviation of 250-m, characterised by an annual precipitation <5,664-mm and slope curvatures less than −0.24. P-values at each node are from a Monte Carlo randomisation test; in order for a split to occur $p$ must be less than 0.05.

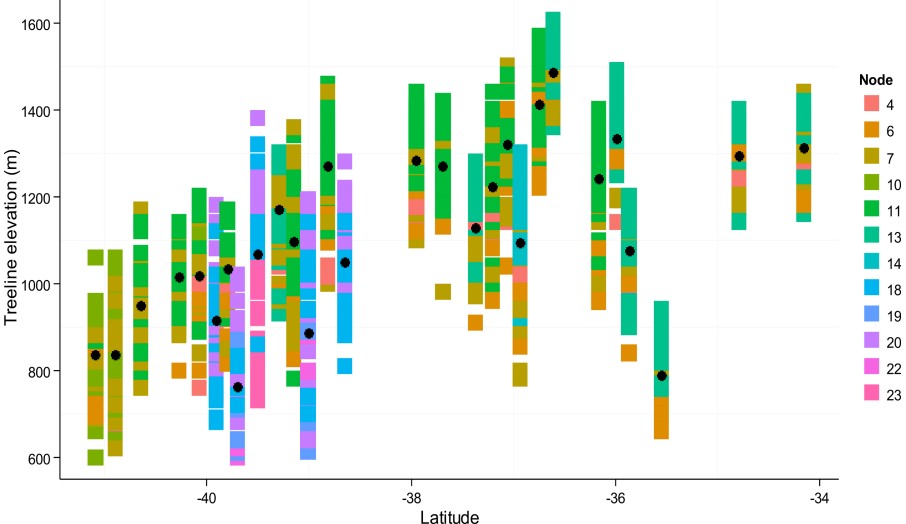

**Figure 7 Tile graph of the distribution of terminal nodes from the conditional inference tree for all sampled treeline locations by site latitude and elevation.** The terminal inference tree nodes represent discrete recursive partitioning pathways in the conditional inference tree (see Fig. 6). Black points represent the mean treeline elevation at each study area.

*Walsh et al., 2003*; *Bader & Ruijten, 2008*; *Case & Duncan, 2014*), the present study is the first to use model-generated meteorological data, and derived topoclimatic indices, to explain treeline variation. Hourly meteorological data generated by the TAPM model enabled the investigation of variability in a number of factors influencing treelines that are notoriously hard to quantify, particularly at local spatial scales in mountain environments. Spatially-explicit data for variables such as minimum or maximum daily temperatures, long-wave radiation, and relative humidity are key variables relevant to characterising physiological stress at treeline but, in previous studies, have been typically limited to collec-

tion by on-site data loggers. Thus, our study has demonstrated potential for the application of such methods to ecological research in mountainous areas. Modelling results indicate that half of the variation in treeline elevation deviations can be explained by a mixture of local landform and topoclimatic factors and mean regional effects of precipitation and growing season temperature. This result corroborates the idea that treeline position is driven by effects that occur at multiple spatial scales, where regional-scale climates can act to modulate or constrain local scale processes and their effects on treelines (*Daniels & Veblen, 2003*; *Elliott & Kipfmueller, 2011*; *Case & Duncan, 2014*). Further, the ways in which the explanatory variables predict treeline elevation deviations across our study areas appear to be geographically-structured, reinforcing the idea that distinct treeline drivers are associated with particular mountain regions (*Case & Hale, 2015*).

Clear topoclimatic gradients were identified using TAPM-generated meteorological data, suggesting that there is a characteristic set of topographically-mediated processes acting across these treelines in New Zealand. Further, different treeline study areas were grouped by distinct topoclimatic conditions as represented by these gradients, likely reflecting the way in which geographic differences in mountain characteristics and regional climates drive these patterns. It is well recognised that the size of mountain ranges and their orientation relative to prevailing winds, valley widths and slope angles all affect valley-scale thermodynamic processes that regulate wind speeds, temperature extremes, and atmospheric moisture levels at different locations (*Sturman & Tapper, 2006*). From a treeline research perspective, the ability to reliably identify and classify treeline study sites at more local scales in terms of their topoclimatic characteristics represents an ongoing gap in treeline research (*Malanson et al., 2011*), and our approach therefore shows promise in this regard. Further verification, using both field work and modelling, of the types of topoclimatic scenarios illuminated in this study would shed further light on particular aspects of topoclimatic effects at treelines.

Most of the treeline locations investigated in this study were positioned between 100 and 300 m below the potential study area maxima, suggesting that southern beech treelines at many locations across New Zealand are likely occurring well below their temperature-based limit. A recent GIS-based analysis of New Zealand beech treeline elevations (*Case & Duncan, 2014*) showed that there was considerable country-wide variation in beech treeline position across the country. At a more local scale, *Wardle*'s (*1985a*; *1985b*; *1985c*) observations of beech treelines in the South Island's Craigieburn Range suggested that treelines in this region are locally-depressed in valley heads and gullies and in other situations where fire had cause past removal of forest. However, his estimates placed these treelines 100 m lower, at most, than what he considered to be climatic treeline. Clearly, based on results from the present study, beech treeline positions in New Zealand are more locally-variable than previously described. Certain areas of the country in particular, such as along the spine of the lower Southern Alps, display highly depressed treeline locations. On the whole, these results highlight the benefits of carrying out analyses of treeline features over large spatial extents in order to provide an accurate characterisation of treeline variability.

Treelines at sites located away from the wettest areas of New Zealand with relatively high mean growing season temperatures (>11.5-°C) were more likely to be situated closer to their maximum potential elevation. Variation in these factors essentially describes differences in regional thermal regimes across the 28 study areas; the effect of thermal regime on treeline position in this study is consistent with previous research that has shown that warmer, drier regions have higher treelines (*Case & Duncan, 2014*). At more local scales, results also indicate that New Zealand's beech treelines are, indeed, higher and closer to their potential maximum growth limits on relatively steep, convex landform positions, consistent with *Wardle*'s (*1985c*) observations. This effect is contrary, however, to what is observed at many northern-hemisphere treelines, where more exposed convex sites have been shown to be typically unfavourable for tree establishment (*Holtmeier, 2009*). In the complex topography of New Zealand's mountains, ridge-to-valley gully features occur regularly along valley sides and may act to channel slope-scale air movement (*Sturman & Tapper, 2006*), thus enhancing the detrimental effects mentioned above. Concave landform situations may also be indicative of where recurring avalanche and landslide disturbances occur. Thus, it is possible that southern beech species are able to reach higher elevations on convex, steeper slopes due to the more stable atmospheric and geomorphic conditions in these locations. Nonetheless, our results also indicate that extremely steep slope gradients (>100%) are associated with depressed treelines, likely a reflection of the physical constraints on tree establishment at such locations (*Macias-Fauria & Johnson, 2013*).

All topoclimatic stressors and ameliorators, as defined by our topoclimatic indices, were relatively highly important in explaining treeline deviations. In general, treelines deviated increasingly further downhill from their potential maximum elevations with increasing topoclimatic stress (i.e., desiccation, photoinhibition, and frost) but were closer to their maxima at locations with higher solar radiation input. The night-time movement of cold air from upper to lower elevations will increase the potential for frost and early morning photoinhibition, while strong daytime upslope wind movement will likely enhance desiccation conditions in both summer and winter. There will therefore likely be a combined effect of these stressors in certain locations at elevations lower than the maximum potential treeline. Cumulative effects can act across seasons; for instance, early summer frosts may disrupt the de-hardening of leaf tissues thereby exacerbating desiccation damage during the following winter (*Cochrane & Slatyer, 1988*). These types of stresses will act to maintain treelines locally at lower elevations, limiting their advance, despite possible warmer mean temperatures relative to higher elevations. Our results are also in line with field-based evidence of the detrimental effects of photoinhibition, desiccation, and frost on southern beech seedlings (*Sakai & Wardle, 1973*; *Wardle, 1985a*; *Wardle, 1985b*; *Greer, Wardle & Buxton, 1989*; *Ball, Hodges & Laughlin, 1991*; *Harsch, 2010*), and lend support to the general hypothesis that abrupt treeline boundaries form at elevations where strong effects of physiological stressors on seedling establishment override the more gradual effect of decreasing temperature with increasing elevation (*Harsch & Bader, 2011*). Further, the positive effect of solar radiation illustrated in our results has been also noted by a number of studies (*Danby & Hik, 2007a*; *Danby & Hik,*

*2007b*; *Elliott & Kipfmueller, 2010*; *Case & Duncan, 2014*), although this was not the case in others (*Körner & Paulsen, 2004*; *Treml & Banaš, 2008*). Higher summer insolation will generally lead to more favourable growing conditions and, thus, produce treelines that are closer to their climatically-driven maximum elevation.

It is well-recognised that treeline patterns are driven, in certain contexts, by non-linear, threshold-like responses to underlying environmental factors (*Malanson, 2001*; *Danby & Hik, 2007b*; *Harsch & Bader, 2011*). In this study there were clear threshold-type responses in the way treeline elevations deviated from their maxima in relation to many of the explanatory variables. For instance, treelines generally deviated further from their potential maximum elevations below a growing season temperature of about 11.5 °C, consistent with findings of *Cieraad & McGlone (2014)*. Similarly, there was a clear and relatively abrupt shift of treelines toward higher elevations as terrain shape switched in form from concave to flat and then to convex. Further, threshold-like responses of treelines to several of the topoclimatic indices, including January and July photoinhibition and frost indices and July insolation, were also evident. On the whole, our results provide further evidence that specific levels of a factor, or sets of factors, can invoke abrupt responses in the physiological (e.g., carbon allocation), demographic (e.g., recruitment) and/or ecological (e.g., competition) mechanisms that underpin treeline formation.

Fifty percent of the variation in treeline elevation deviation could not be explained by our models; explanatory factors appeared to be most useful in predicting deviations of 300-m or less. It is highly likely that treeline elevation deviations of more than 300-m were driven by local disturbances that were not well-represented by our explanatory factors. The effects of disturbances, including those from fire (*Ledgard & Davis, 2004*), heavy winds (*Martin & Ogden, 2006*) and snowfalls (*Wardle & Allen, 1984*), and tectonic activity (*Allen, Bellingham & Wiser, 1999*; *Haase, 1999*; *Vittoz, Steward & Duncan, 2001*) are widespread and significant throughout southern beech forests and are apparent throughout many treeline zones (*Wardle, 2008*; *Case & Hale, 2015*). Further, southern beech species in New Zealand are generally slow to recolonise areas after removal, even in lower-elevation forests, due to strong competition with other species (*Wiser, Allen & Platt, 1997*). Thus, disturbance may be a confounding factor at many treeline sites in New Zealand and, without better datasets characterising the spatial distribution of local-scale disturbances across the country, it may be difficult to disentangle their effects from those due to climate. Further, finer scale microhabitat and microclimate effects on ecological interactions are likely also very important in allowing beech seedlings to establish above current positions (*Harsch et al., 2012*). In the case of abrupt treelines, positive feedback processes, where established trees facilitate the recruitment of nearby seedlings through environmental modification, are critical in enabling these treelines to advance (*Wiegand et al., 2006*).

## CONCLUSIONS

This study has demonstrated that models such as TAPM have potential for enabling local-scale investigations of topoclimatic effects at treeline. Clearly, two of the biggest advantages are the generation of spatially-explicit data useful for characterising stress-related effects

and the ability to produce these data at any location. However, it is important to recognise that meteorological data for one year may not be representative of typical topoclimatic conditions occurring at a given location, and it would therefore be useful to average model data over longer periods and for other critical parts of the year such as late spring and early autumn. Overall, we show that landform, topoclimatic, and regional factors together can explain half the variation in local treeline elevation variation and that their influences are geographically-structured and typically non-linear in nature. Ultimately, results from this study could be used to characterise sites with different topoclimatic situations where investigations of local scale microclimate and biotic interactions could be investigated.

## ACKNOWLEDGEMENTS

We thank Richard Duncan, Roddy Hale, Timothy Curran, Peyman Zawar-Reza and the Lincoln University Spatial Ecology Lab Group for reading and/or discussing earlier versions of this work. We also thank the three reviewers whose useful comments greatly improved this manuscript.

### Funding

Funding was provided by Lincoln University. The funders had no role in study design, data collection and analysis, decision to publish, or preparation of the manuscript.

### Grant Disclosures

The following grant information was disclosed by the authors:
Lincoln University.

### Competing Interests

Hannah L. Buckley is an Academic Editor for PeerJ.

### Author Contributions

- Bradley S. Case conceived and designed the experiments, performed the experiments, analyzed the data, contributed reagents/materials/analysis tools, wrote the paper, prepared figures and/or tables, reviewed drafts of the paper.
- Hannah L. Buckley performed the experiments, wrote the paper, prepared figures and/or tables, reviewed drafts of the paper.

### Supplemental Information

Supplemental information for this article can be found online at http://dx.doi.org/10.7717/peerj.1334#supplemental-information.

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
