# Peer review of "Local-scale topoclimate effects on treeline elevations: a country-wide investigation of New Zealand’s southern beech treelines"

_PeerJ, doi:10.7717/peerj.1334_

## Round 0.1 · original submission · Major Revisions

This manuscript has now undergone three reviews and all reviewers were fairly unanimous in their enthusiasm regarding the suitability of this paper for this journal. However, all three reviewers find fault with various parts of the presentation (e.g., the figures and parts of the discussion). One reviewer also expressed concerns regarding the statistical design and methodology (e.g., why only meteorological data from one year?). Detailed suggestions for revision are included here that should provide ample guidance to revising your manuscript.

Reviewer 1 ·

Basic reporting

No comments.

Experimental design

See my general comments for the author.

Validity of the findings

No comments.

Additional comments

General comments:

This paper presents the results of a modelling analysis of treeline elevation in New Zealand. The authors estimated treeline elevation using a random forest and conditional inference tree modelling approach, using 17 predictor variables including landform, environmental, meteorological, and topoclimatic indices. The key findings were:

1. Curvature (i.e., convexity and concavity) was the strongest predictor of treeline elevation.
2. Local landform and topoclimatic effects were typically more important than regional-scale factors in determining treeline elevation.

This paper would make a strong contribution to literature on treeline dynamics; especially considering the nature of shifting range limits of vegetation in a rapidly changing climate is a timely area of research. The rationale for the study is clear and the introduction is well written and focused. However, before I can recommend this article for publication in PeerJ, I have a concern with the methods that needs to be addressed.

Using weather data from only 2002 poses some challenges that need further explanation. For example, we have no indication of how old these trees are, but suffice to say, they most likely germinated before 2002. Therefore, how can their current distribution be modeled using such recent data? Moreover, how do these values compare with longer-term data, such as the 20th century mean (i.e., within the context of longer term variability)? These data are also drawn from a period of unprecedented change worldwide (e.g., post 1950), so how representative are they? Relying on such a short timeframe to (accurately) explain the result of demographic processes responsible for overall forest extent also ignores what Szeicz and MacDonald (1995) and Danby and Hik (2007) suggest is needed for successful tree establishment in marginal landscapes (e.g., 30-50 years of favorable climate conditions from the time of germination). This timeframe also excludes important inter-annual variability that is undoubtedly inherent to the climate variables selected. The authors appear aware of these limitations, though don’t comment on them until the final paragraph of the manuscript.

The authors either need to provide a strong justification for the time period of their meteorological data or re-run their analyses using meteorological and topoclimatic indices from a longer, more ecologically significant time period.

Specific comments:

l. 108–110: It should be mentioned here that Nothofagus treelines in the study areas are typically abrupt. Without this forewarning, I would find the idea of delineating treeline, which is notoriously gradated in most areas, as a “line” using polygon topology exceedingly dubious.

l. 110: The year is missing for the Case and Duncan reference.

l. 171–172: What was the rationale behind choosing meteorology during these two months and not longer periods during the year? Perhaps warm (>0°C) and cold (<0°C) seasons might be more ecologically significant? Also see general comments above.

l. 187–188: Same comment as above. I feel a stronger rationale is necessary to justify the selection.

l. 236: “…Tsite_max…”

l. 277: This citation should be Cutler et al. 2007 and the reference in the bibliography should be changed to include the remaining four authors.

l. 319: Please explain the rationale for focusing on the top 12 of the 17 predictor variables.

l. 355: I feel this statement is misleading as there have been numerous studies which have used model-generated climate data (e.g., CRU, GISS, SNAP) and topoclimatic variables to explain treeline elevation—either directly or indirectly. It may be true that this is the first study to derive climate and topoclimatic indices explicitly to explain treeline elevation, but there have been other studies that have used comparable metrics to evaluate treeline position (see references below). I feel the authors’ statement needs to be softened by including further context for the novelty of their study, based on further references to the literature.

Allen, T.R., and Walsh, S.J. 1996. Spatial and compositional pattern of alpine treeline, Glacier National Park, Montana. Photogrammetric Engineering and Remote Sensing 62(11): 1261–1268.

Brown, D.G. 1994. Predicting vegetation at treeline using topography and biophysical disturbance variables. Journal of Vegetation Science 5(5): 641-656.

Calef, M.P., McGuire, A.D., Epstein, H.E., Rupp, T.S., and Shugart, H.H. 2005. Analysis of vegetation distribution in Interior Alaska and sensitivity to climate change using a logistic regression approach. Journal of Biogeography 32(5): 863-878.

Mamet, S., Cairns, D., Brook, R., and Kershaw, G.P. 2015. Modeling the spatial distribution of subarctic forest in northern Manitoba using GIS-based terrain and climate data. Physical Geography 36(2): 93–112. doi: 10.1080/02723646.2014.994253.

Paulsen, J., and Körner, C. 2014. A climate-based model to predict potential treeline position around the globe. Alp Botany 124(1): 1–12.

Figure 1: I would suggest adding lat/lon tics to the main map to help readers determine more precisely where the research sites were.

Figure 3: Caption—“Dots represent positions of the 28 study areas in ordination space, coloured by their latitudinal position.” Also, the caption refers to Table 2, which should actually be Table 1.

Figure 5: Please provide labels for the y-axis.

Table 1: Please provide the rationale for choosing the 0.25 loading threshold. “Raindays” should be “rain days”.

·

Basic reporting

The manuscript is very well referenced and reports in detail the state of the art and the research methodology.

Experimental design

The research questions are clearly stated and the methodology is explained in detail. The experimental design is rigorous, correct and conducted to a high technical standard.

Validity of the findings

No comments.

Additional comments

I am impressed with the methodology employed to disentangle factors that set treeline in the study area. The amount of detail provided in the introduction and methods sections is notable and welcome and the study design has the potential to set new standards in treeline research, reason for which I wholeheartedly recommend it for publication.
However I have a major concern which relates to the lack of a concise discussion, with crisp clear messages to the point. I feel the authors reiterate similar content while loosing the interest of the reader and do not spend enough time on addressing properly the main findings. In particular the nonlinearity of responses to environmental variables would deserve a more in depth discussion. As well, some of the figures (e.g. Figure 3 and 4) could be passed to Supplementary material. Additionally, a better use of colours in Figure 3 would ease understanding of geographical and ecological gradients of interest (e.g., using gradients instead of rainbow).
I recommend the authors revise this section to address better the main findings and bring the take-home messages within easy reach.

Reviewer 3 ·

Basic reporting

no comments

Experimental design

article has appropriate methods

Validity of the findings

findings valid

Additional comments

This study finds local treelines are affected by multiple variables and these influences vary from region to region. I have no issues with the methods or the interpretation. My only comment is that I would like to see an analysis of the variability over the full 28 sites adjusted for latitude.
Minor comments
Line 46-48 “effects” is used three times in the one sentence
Line 117. These latitudes are incorrect. Your northern site is approximately 39o S and your southern site is approx. 46o S, not 34o and 42o.
Line 217-218. Relative humidity data mentioned twice. And it should be wind speed not winds speed.
Line 220 some terms such as WShourly are not defined
Line 225. What is the difference between Ntotal and Ndaytime? This is unclear.
Line 308-309. Sentence incomplete.
Line 319-344. Talking only in terms of deviations makes this section readable than if you were to talk about lower or higher elevations of treeline associations.
Figure 3. There are so many colours and vectors these figures are difficult to interpret.

---

## Round 0.2 · accepted · Accept

The authors have done a very thorough job with the revisions. Figure 1 has a typo (I believe the Y axis level should be 'deviations').